# Contrastive Representations for Temporal Reasoning

**Alicja Ziarko**[1 2 3]     **Michał Bortkiewicz**[4]     **Michał Zawalski**[1, 6]
**Benjamin Eysenbach**[5 †]     **Piotr Miłoś**[1 3 †]
[1]University of Warsaw     [2]IDEAS NCBR     [3]IMPAN
[4]Warsaw University of Technology     [5]Princeton University     [6]NVIDIA
alicja.ziarko@uw.edu.pl

## Abstract

In classical AI, perception relies on learning state-based representations, while planning — temporal reasoning over action sequences — is typically achieved through search. We study whether such reasoning can instead emerge from representations that capture both perceptual and temporal structure. We show that standard temporal contrastive learning, despite its popularity, often fails to capture temporal structure due to its reliance on spurious features. To address this, we introduce **Contrastive Representations for Temporal Reasoning** (CRTR), a method that uses a negative sampling scheme to provably remove these spurious features and facilitate temporal reasoning. CRTR achieves strong results on domains with complex temporal structure, such as Sokoban and Rubik's Cube. In particular, for the Rubik's Cube, CRTR learns representations that generalize across all initial states and allow it to solve the puzzle using fewer search steps than BestFS — though with longer solutions. To our knowledge, this is the first method that efficiently solves arbitrary Cube states using only learned representations, without relying on an external search algorithm.

## 1 Introduction

Machine learning has achieved remarkable progress in vision [19], control [8], and language [23, 11]. Yet it still struggles with structured, combinatorial reasoning. Even simple tasks like planning in puzzles or verifying symbolic constraints remain difficult for end-to-end systems [22, 14]. State-of-the-art solvers rely on computationally expensive search methods such as A* or BestFS [10]. This work asks: *Can we learn representations that reduce or eliminate search in combinatorial reasoning?*

We study whether temporal contrastive learning [17, 8] can enable efficient reasoning directly in latent space. While contrastive learning has shown promise in control, its performance in combinatorial domains is limited. We identify a key failure mode: embeddings overfit to instance-specific context rather than temporal dynamics.

We introduce Contrastive Representations for Temporal Reasoning (CRTR), a simple, theoretically grounded contrastive method that uses in-trajectory negatives. By distinguishing temporally distant states within the *same* episode, CRTR avoids reliance on irrelevant context and instead encodes meaningful temporal dynamics.

Our main contributions are:

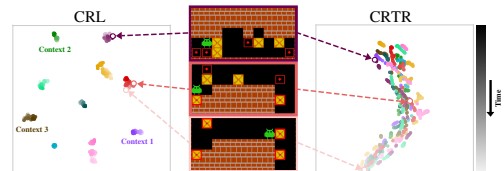

Figure 1: **CRTR learns temporally structured representations.** t-SNE visualization of Sokoban embeddings learned by CRL (left) and CRTR (right). CRL clusters within trajectories, missing global structure. CRTR organizes embeddings across trajectories and time (vertical axis), capturing dynamics essential for planning.

Preprint.

1. We identify a critical failure mode of contrastive learning in domains with complex temporal structure.

2. We propose Contrastive Representations for Temporal Reasoning (CRTR), a novel, theoretically grounded method using in-trajectory negatives to learn temporally structured representations.

3. We show that CRTR outperforms prior methods on 4 of 5 combinatorial reasoning tasks, and enables solving the Rubik's Cube with fewer search steps than BestFS (though with longer solutions).

## 2   Method

**Failure of naïve Contrastive Reinforcement Learning in combinatorial domains.**   A straightforward approach to learning representations $\phi(s)$ is to employ contrastive reinforcement learning (CRL) [8]. We applied this method in Sokoban — a puzzle game where an agent must push boxes to target locations in a maze. Each problem instance is generated with a random wall pattern. Fig. 1 shows a t-SNE projection of representations learned by contrastive reinforcement learning on this task. The representations from standard CRL primarily encode the layout of the walls and not the temporal structure of the task. The reason representations use those features is that doing so minimizes the contrastive objective. Each batch element typically comes from a different maze, so representations that use the wall pattern to detect positive vs negative pairs achieve nearly perfect accuracy.

**A mathematical explanation.**   The failure of temporal contrastive learning can be explained by the presence of a *context* variable $c$. Each trajectory $\tau = (s_1, \ldots, s_T)$ can be decomposed into a fixed context $c$ (e.g., the wall and goal layout in Sokoban) and a temporal part $(f_1, \ldots, f_T)$ that evolves over time (e.g., player and box positions). For the sake of theoretial analysis, we assume that for any $i < j$, the future state $s_j$ is conditionally independent of $c$ given $s_i$ ($s_j \perp c \mid s_i$), which holds in Sokoban.

**Learning representations that ignore context: an idealized algorithm.**   Our method samples negatives $(x, x_-)$ that share the same context, so context features cannot help distinguish positives from negatives and are excluded from the learned representations. Formally, we draw $c \sim \mathcal{P}(C)$, positives $(x, x_+) \sim \mathcal{P}(X, X_+ \mid c)$, and negatives $x_-^{(i)} \sim \mathcal{P}(X \mid c)$. The objective is $\max_f \mathcal{L}(f) \triangleq \mathbb{E}\left[\frac{1}{N} \sum_{j=1}^{N} \frac{e^{f(x_j, x_{j+})}}{e^{f(x_j, x_{j+})} + \sum_{k=1}^{N-1} e^{f(x_j, x_{j-}^k)}}\right]$ a lower bound on $I(X; X_+ \mid C)$ [15]. Using the decomposition $I(X_+; X \mid C) = I(X_+; X) - I(X_+; C)$   (since $X_+ \perp C \mid X$), we see that the objective encourages maximizing temporal information $I(X; X_+)$ while minimizing context information $I(X_+; C)$.

**A practical method.**   While the idealized method is useful for analysis, it assumes that the context is clearly separable from the observation, which is rarely the case. We propose a practical algorithm that avoids this assumption. The method modifies contrastive sampling: instead of one positive per trajectory, we sample multiple positives, so that some negatives in the batch come from the same trajectory at different times. Implementing this idea in practice requires changing just a few lines of code from prior temporal contrastive learning methods, as highlighted in Appendix I. Using data sampled in this way guarantees that some negative training pairs in each batch come from the same trajectory. We compare with potential alternative approaches in Appendix J. These within-trajectory negatives differ systematically from positives and push the model to focus on temporal variations rather than trajectory-wide constants. This method can be applied without any knowledge of the context, even to problems without a constant context (e.g., the Rubik's Cube).

## 3   Experiments

**Experimental setup.**   We evaluate on five combinatorial reasoning tasks: Sokoban [7], Rubik's Cube, N-Puzzle [12], Lights Out [2], and Digit Jumper [3]. Most of these are NP-hard [6, 4, 20] and serve as standard RL benchmarks [1, 18, 25]. See Appendix A for full environment details.

Baselines include: standard CRL [21, 17, 8], a supervised value-based approach[5, 24]; Deep-CubeA [1]; and a random network. We test with and without search. When we use search, all methods, including DeepCubeA, use BestFS for planning. In the setting without search we plan

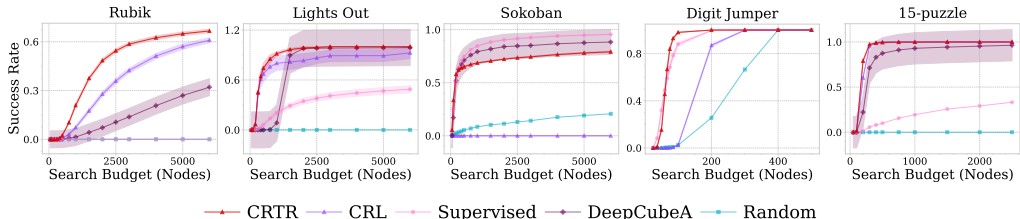

Figure 2: **CRTR performs well in all the evaluated domains.** Success rate as a function of search budget across five domains. CRTR compared to baselines: CRL [8], Supervised [5] and DeepCubeA [1]. Results are averaged over 5 seeds; shaded regions indicate standard error.

by greedily selecting the neighbor with minimum predicted distance under known, deterministic dynamics. All the methods avoid loops by only considering states that were not already processed. Further evaluation details are provided in Appendix D. The hyperparameters for each method are provided in Appendix C.

**Context-free representations for combinatorial reasoning.** We analyze learned representations in Sokoban, where wall layouts provide clear context features. Fig. 1 compares CRTR with standard temporal contrastive learning (CRL). Using t-SNE, we find that CRL clusters trajectories by static context, encoding all states from a trajectory similarly, while CRTR aligns states by temporal progress, indicating that it discards irrelevant context in favor of task-relevant structure.

Our second experiment studies whether the learned CRTR representations are useful for decision making and how they compare to supervised approaches [1, 5]. We use the representations to construct a heuristic for search. As shown in Fig. 2, CRTR consistently achieves among the highest success rates, strictly the best in two of them. The strong performance relative to CRL highlights the importance of removing context information from learned representations. In Appendix E, we provide additional, smaller-scale experiments showing that these improvements also hold when using a non-greedy solver. The improvement in performance in comparison to supervised baselines suggests that CRTR's advantage comes from representing values as distances between learned representations rather than as outputs of a monolithic neural network.

The t-SNE visualizations (Figure 1) suggests that CRL focuses primarily on the static context, while CRTR focuses on the temporal structure. Below, we present additional empirical evidence supporting this interpretation.

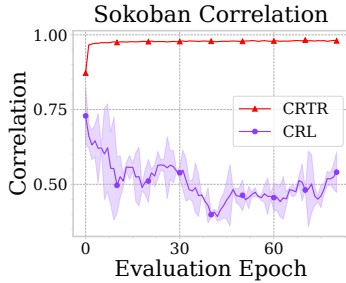

Figure 3: **Distances given by CRTR representations reflect the temporal structure well.** Correlation (Spearman's $\rho$) between the distance induced by learned embeddings and actual distance across the training, CRTR compared with CRL.

We perform further analysis in Sokoban environments. Without negative pairs, the classification task becomes nearly trivial: the model leverages context cues to achieve close to 100% accuracy (Appendix E). Despite this, the learned representations exhibit low correlation with ground-truth state-space distances (Figure 3), indicating that the model ignores temporal structure and instead relies on static context. In contrast, CRTR prevents reliance on contextual shortcuts, resulting in representations that better capture the underlying geometry of the environment (Figure 3). We provide a similar analysis for Digit Jumper in Appendix E. We also demonstrate that using CRTR leads to improved temporal structure in robotic domains (See Appendix F). In Appendix H we show that CRTR results in representations that optimize conditional mutual information $I(X, X_+|C)$, while CRL does not.

**Is search necessary?** Do good representations allow us to solve combinatorial problems without search, or at least reduce the amount of search required to get high success rates? We study this question by using the learned representations to perform greedy planning for up to 6000 search steps.

We present the results from this experiment in Figure 4, showing the fraction of problems solved with fewer than a certain number of steps. We compare to the variant of CRTR used in Sec. 3. On 4 / 5 tasks, CRTR solves nearly all problem instances. The key takeaway is thus: *for most problems,*

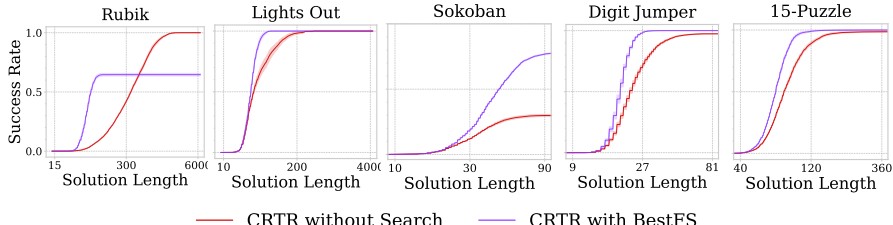

Figure 4: **CRTR solves most tasks without requiring any search.** We plot the fraction of configurations solved with a solution length of at most $x$, while limiting the number of nodes created to 6000. Surprisingly, on the Rubik's cube CRTR achieves a higher success rate *without* search, solving all board configurations within the budget.

*CRTR can find solutions without needing any search at all.* Perhaps the most interesting result is the Rubik's cube, where we found that our representations can solve all problem instances in less than 6000 moves. Surprisingly, using search decreases the total fraction of Cube configurations that are solved. However, avoiding search comes at a cost: the solutions found without search are typically longer than those found with search.

This simple greedy approach — just picking the neighbor closest to the goal — starts to show hints of algorithmic behavior. On Rubik's Cube, for example, it learns something that looks like a rudimentary form of block-building (See Fig. 5), a common strategy used by humans for solving the cube. This block building strategy was not programmed or explicitly rewarded, but instead emerged from training the representations on random data.

**Ablation experiments.** Appendix J presents additional ablation experiments. We find that *(1)* our strategy for sampling data (Alg. 2) outperforms several alternatives, and *(2)* CRTR is robust to the `repetition_factor` hyperparameter, with 2 being a good choice in all settings we have tested.

## 4  Conclusions

In our work, we introduced CRTR, an algorithm for learning high-quality representations in combinatorial reasoning tasks. Our analysis revealed a critical limitation of prior approaches: when training demonstrations are separable, their learned representations become trivial and ineffective for planning. CRTR addresses this by balancing global negatives, which capture overall task structure, with local negatives, which enforce temporal consistency. Experimental results across five domains highlight its effectiveness. Notably, the representations learned with CRTR can successfully guide search even without explicit planning, suggesting a promising direction for future research. and broad applicability. We share the code for reproducibility. [1]

---

[1]Our code is available at: https://github.com/combinatorialreasoning/crcr.

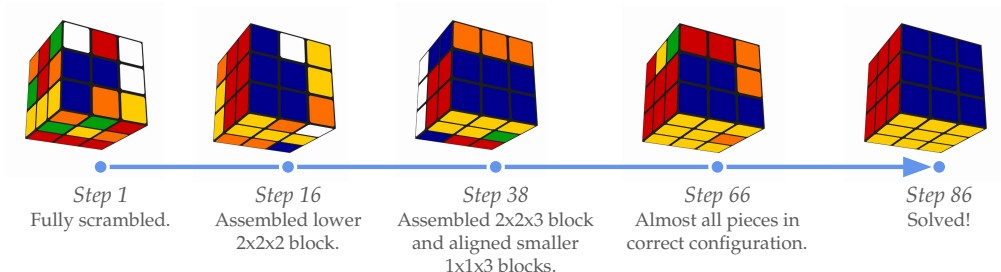

| Step 1 | Step 16 | Step 38 | Step 66 | Step 86 |
| Fully scrambled. | Assembled lower 2x2x2 block. | Assembled 2x2x3 block and aligned smaller 1x1x3 blocks. | Almost all pieces in correct configuration. | Solved! |

Figure 5: **CRTR without search exhibits a block-building-like behavior.** Intermediate states from solving a randomly scrambled cube, illustrating how the algorithm gradually builds partial structure. The average solve is about 400 moves, and we see similar block building behavior across solves.

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

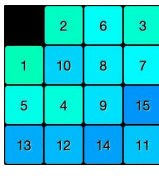
(a) N-Puzzle.

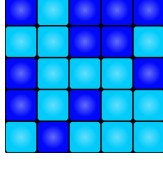
(b) Lights Out.

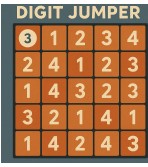
(c) Digit Jumper.

Figure 6: **Environments:** Our experiments used Sokoban (Fig. 1), the Rubik's Cube (Fig. 5), and the three environments shown above.

## A Environments

**Sokoban.** Sokoban is a well-known puzzle game in which a player pushes boxes onto designated goal positions within a confined grid. It is known to be hard from a computational complexity perspective. Solving it requires reasoning over a vast number of possible move sequences, making it a standard benchmark for both classical planning algorithms and modern deep learning approaches [7]. Solving Sokoban requires balancing efficient search with long-term planning. In our experiments, we use 12×12 boards with four boxes.

**Rubik's Cube.** The Rubik's Cube is a 3D combinatorial puzzle with over $4.3 \times 10^{19}$ possible configurations, making it an ideal testbed for algorithms tackling massive search spaces. Solving the Rubik's Cube requires sophisticated reasoning and planning, as well as the ability to efficiently navigate high-dimensional state spaces. Recent advances in using neural networks for solving this puzzle, such as [1], highlight the potential of deep learning in handling such computationally challenging tasks.

**N-Puzzle.** N-Puzzle is a sliding-tile puzzle with variants such as the 8-puzzle (3×3 grid), 15-puzzle (4×4 grid), and 24-puzzle (5×5 grid). The objective is to rearrange tiles into a predefined order by sliding them into an empty space. It serves as a classic benchmark for testing the planning and search efficiency of algorithms. The problem's difficulty increases with puzzle size, requiring effective heuristics for solving larger instances.

**Lights Out.** Lights Out is a single-player game invented in 1995. It is a grid-based game in which each cell (or *light*) can be either on or off. Pressing a cell flips its state and those of its immediate neighbors (above, below, left, and right). Corner and edge lights have fewer neighbors and therefore affect fewer lights. The goal is to press the lights in a strategic order to turn off all the lights on the grid.

**Digit Jumper.** Digit Jumper is a grid-based game in which the objective is to get from the top-left corner of the board to the bottom-right corner. At each point, the player can move $n$ steps to the left, right, up, or down, where $n$ is determined by the number written on the current cell. *Digit Jumper* is an example of an environment with a constant context, as is *Sokoban*.

## B Best-First Search

Best-First Search (BestFS) greedily prioritizes node expansions with the highest heuristic estimates, aiming to follow paths that are likely to reach the goal. Although it does not guarantee optimality, BestFS offers a simple and efficient strategy for navigating complex search spaces. The high-level pseudocode for BestFS is presented in Algorithm 1.

---
**Algorithm 1** Best-First Search [10]

---
**while** has nodes to expand **do**
    Take node $N$ with the highest value
    Select children $n_i$ of $N$
    Compute values $v_i$ for the children
    Add $(n_i, v_i)$ to the search tree
**end while**

---

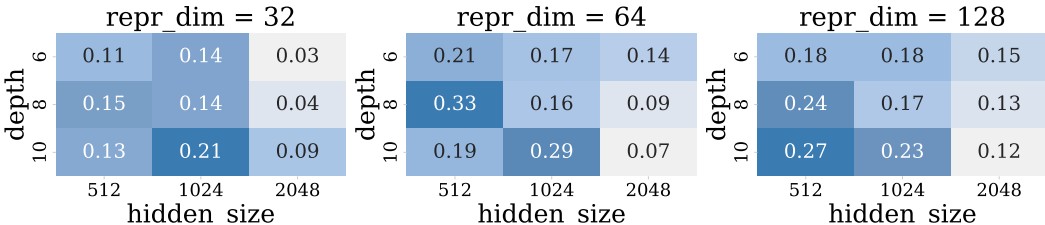

Figure 7: Grid of network's depth, representation dimension and hidden dimension. The success rate is evaluated on cubes scrambled with 10 random moves.

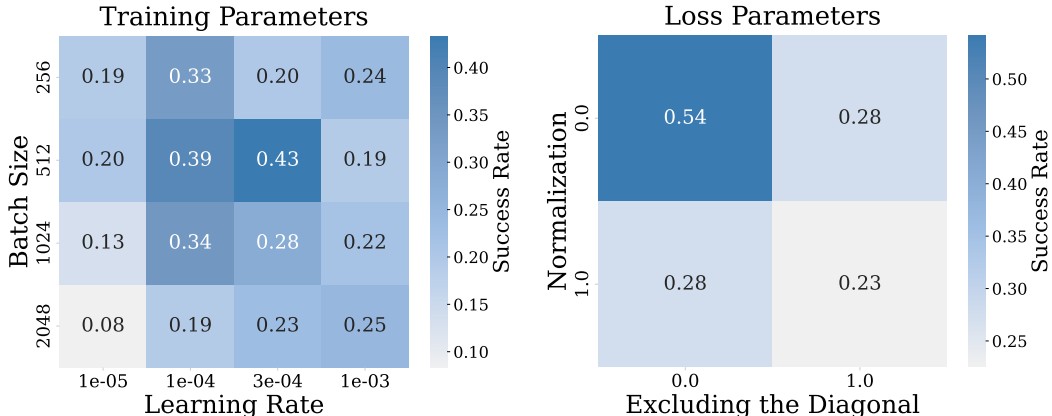

Figure 8: Learning rate and batch size grid for Rubik's Cube. The success rate is evaluated on cubes scrambled with 10 random moves after 700k training steps.

Figure 9: **CRTR is the only effective normalization strategy in Sokoban.** Effect of using negatives in contrastive learning in Sokoban. We compare the setting where the distance to positives is normalized by the sum over all batch elements or only the in-batch negatives. The success rate is evaluated on cubes scrambled with 10 random moves after 400k training steps.

## C  Training Details

Code to reproduce all results is available in the anonymous repository referenced in the main text. Below, we document the training procedures for the supervised baseline, contrastive baseline, and CRTR.

**Training data.**  For Sokoban, we use trajectories provided by Czechowski et al. [5] and train on a dataset of $10^5$ trajectories. For 15-Puzzle, Rubik's Cube, and Lights Out, we generate training trajectories by applying a policy that performs $n$ random actions, where $n$ is set to 150, 21, and 49, respectively. In the case of 15-Puzzle, we additionally remove single-step cycles from the dataset to improve data efficiency. For Digit Jumper, we generate training data by sampling a random path from the upper-left corner to the bottom-right corner on a standard $20 \times 20$ grid. All grid cells not required for this path are filled by sampling uniformly from the set $1, \ldots, 6$. The network for Digit Jumper typically converges after a few hours of training, so we train until convergence is observed. For Sokoban, Rubik's Cube, Lights Out, and 15-Puzzle, we adopt an unlimited data setup and train all models for two days. This results in the models performing approximately $8 \times 10^6$ gradient updates for Rubik's Cube, $7 \times 10^6$ for 15-Puzzle, and $9 \times 10^6$ for Lights Out.

**Training hyperparameters.**  We use the Adam optimizer with a constant learning rate throughout training. A learning rate of $0.0003$ was found to perform well across all environments, with the exception of Lights Out, where this setting led to unstable training. For this environment, we instead use a reduced learning rate of $0.0001$. In all environments, we use a batch size of 512. The choice of

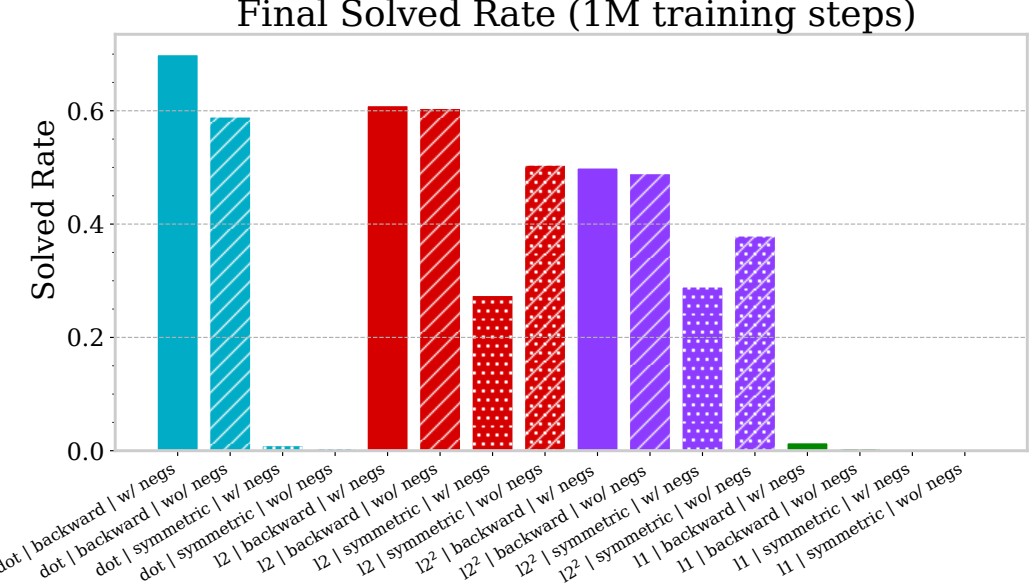

Figure 10: Success rate on Rubik's Cube scrambled with 10 random moves, for models trained with different contrastive losses. Models using the backward loss consistently achieve better performance than those using the symmetric variant. Using the dot product without in-trajectory negatives performs similarly to the $\ell_2$ metric, while combining the dot product with in-trajectory negatives yields the highest success rate. In contrast, combining in-trajectory negatives with symmetric loss results in a drop in performance, likely because, in CRTR, such negatives are often closer to the correct solution in the state-space.

learning rate and batch size was guided by the performance of the contrastive baseline on Rubik's Cube. Specifically, we evaluated solve rates on cubes shuffled 10 times, as shown in Figure 8. We also conducted grid searches to find the optimal training parameters (learning rate and batch size) for the supervised baseline on Sokoban, Lights Out, and Rubik's Cube . We use the same batch size and learning rate across all methods and environments, with the exception of Lights Out, where increasing the batch size and learning rate in the supervised baseline led to a higher success rate.

**Network architecture.** We adopt the network architecture proposed by Nauman et al. [16], using 8 layers with a hidden size of 512 and a representation dimension of 64. This configuration was found to yield optimal performance for the contrastive baseline on Rubik's Cube, as illustrated in Figure 7. We observed that this architecture performs well in all environments except for two cases:

- In Sokoban, a convolutional architecture was required to achieve strong performance.
- In Lights Out, the convolutional network was necessary to ensure training stability.

**Test set.** For Sokoban, we construct a separate test set comprising 100 trajectories, which is used to compute evaluation metrics such as accuracy, correlation, and t-SNE visualizations. For all other environments, a separate test set is unnecessary, as we train for only a single epoch. In this setting, evaluation is performed directly on unseen data sampled during training.

**Contrastive loss.** We use the backward version of the contrastive loss, which we found to consistently outperform the symmetrized variant on Rubik's Cube as shown in Figure 10. We also found the backward version to work better on 15-Puzzle and slightly better in the remaining environments.

For Rubik's Cube, we use the dot product as the similarity metric. Performance across different metrics is presented in Figure 10. While the contrastive baseline performs comparably under the $\ell_2$ metric, CRTR achieves significantly better results with the dot product. Based on similar empirical evaluations, we use the following metrics for other environments:

- Lights Out: $\ell_2$ distance,
- Digit Jumper and 15-Puzzle: dot product,

- Sokoban: squared $\ell_2$ distance.

We set the temperature parameter in the contrastive loss to the square root of the representation dimension.

**Supervised baseline.** The supervised baseline takes as input a pair of states and predicts the distance between them by classifying into discrete bins, where the number of bins corresponds to the maximum trajectory length observed in the dataset.

In all environments, the supervised baseline uses the same architecture as the contrastive baseline.

## D   Evaluation Details

We evaluate all networks on $1000$ problem instances per environment. For Rubik's Cube, each instance is a cube scrambled using $1000$ moves. For 15-Puzzle, Lights Out, and Digit Jumper, evaluation boards are sampled randomly. For Sokoban, we follow the same instance generation procedure as described by Czechowski et al. [5].

## E   Additional Experiments

**A\* solver.** To verify that the improvements achieved by CRTR are not specific to greedy solvers, we conducted an additional experiment using the A\* search algorithm. A\* employs a heuristic function of the form heuristic $+\,\alpha \cdot$ cost, where varying $\alpha$ allows trading off between the search budget required to solve the problem and the average solution length. As shown in Table 1, for the Rubik's Cube, increasing $\alpha$ from $0$ (equivalent to BestFS) to $500$ consistently yields better performance for CRTR compared to CRL. We therefore hypothesize that the improvement reported in Section 3 is not specific to greedy solvers.

Table 1: **CRTR effectiveness is not BestFS specific.** A\* search results on the Rubik's Cube with a node budget of 6000, varying $\alpha$ in the priority function. CRTR performs better than CRL for all values of $\alpha$, achieving shorter solution lengths and higher solved rates.

| $\alpha$ | 0 | 100 | 200 | 300 | 400 |
|---|---|---|---|---|---|
| CRTR Avg. Solution Length | 56.76 | 46.35 | 38.42 | 32.84 | 29.16 |
| CRTR Success Rate | 0.63 | 0.62 | 0.59 | 0.54 | 0.33 |
| CRL Avg. Solution Length | 62.96 | 49.88 | 41.94 | 36.11 | 31.77 |
| CRL Success Rate | 0.54 | 0.50 | 0.44 | 0.40 | 0.30 |

**No-search results.** The no-search approach selects, at each step, the state that appears most likely to lead toward the solution—based on the learned representation. If the representation were perfect, this strategy would yield optimal solutions. In practice, however, suboptimal representations often cause the agent to wander through latent states far from the goal before eventually converging. As a result, the quality of the representation is reflected in the length of these trajectories: the better it captures directionality in latent space, the shorter the resulting solutions.

Table 2 reports the average solution lengths for the no-search approach on Rubik's Cube and 15-Puzzle. The results suggest that the representations learned by CRTR are better suited to this approach than those learned by the contrastive baseline, and they significantly outperform those derived from the supervised method. This supports the conclusion that CRTR provides a more reliable notion of direction in latent space. Notably, the average solution lengths for both CRTR and CRL are shorter than the length of training trajectories in 15-Puzzle (150), indicating evidence of trajectory stitching.

We furthermore present the distributions of solution lengths for all the methods in Figure 11.

**Accuracy in Sokoban training.** During the training of CRL on the Sokoban environment, a perfect accuracy is acquired almost immediately, due to the method relying on the context, as demostrated in Figure 14.

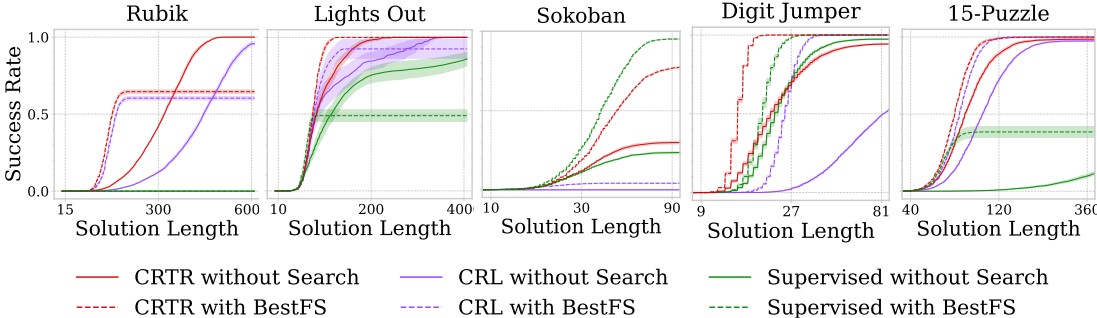

Figure 11: **CRTR produces shorter solutions without explicit search in comparison to baselines. Search can help reduce solution length further.** Fraction of boards solved with a solution length of at most $x$, comparing CRTR to baselines. Figure 4 in the main text presents analogous results, but only CRTR, for clarity.

Table 2: Average solution length of the baselines and CRTR on Rubik's Cube and 15-Puzzle without using search. Supervised baseline fails to solve Rubik's Cube without search.

| Problem | CRTR | Contrastive Baseline | Supervised Baseline |
|---|---|---|---|
| Rubik's Cube | 448.7 | 1830.3 | NaN |
| 15-puzzle | 82.4 | 119.5 | 1054.3 |

**Digit Jumper analysis.** Digit-Jumper is an example of another constant context (defined in Sec. 2) environment, as is Sokoban. It is therefore another environment in which CRL fails rather spectacularly and therefore, we observe a similar effect to that seen in Sokoban when comparing CRTR to standard CRL. As shown in Figure 12, CRL rapidly achieves 100% training accuracy. However, despite this perfect accuracy, the resulting representations exhibit poor correlation with actual temporal structure (Figure 13). This is consistent with the t-SNE visualization (Figure 15): as with Sokoban, CRL collapses each trajectory into a single point in the representation space, discarding temporal information. In contrast, CRTR preserves a clear temporal structure within the latent space (see Figure 15). For non-constant context environments, the difference in representation quality is also visible in success rates, accuracy and correlation, it is however much less pronounced.

## F   Generalization to Temporal Reasoning in Non-Combinatorial Domains

To investigate whether CRTR also identifies temporal features in non-combinatorial domains, we apply it to a dataset of robotic manipulation trajectories (the Adroit dataset from D4RL [9]). Those tasks require using a high-dimensional robotic hand to perform fine motor activities, and are designed to test fine motor control and long-horizon planning. We quantify representation quality by measuring the predicted distance from each state in a trajectory to the final state in a trajectory. Specifically, we look at the rank correlation between the time step and predicted distance, with a correlation of 1 indicating that the learned representations are highly predictive of the temporal distance from each state to the final state.

We look at the correlation through training for CRTR and CRL (Fig. 16). CRTR results in a higher correlation (more than 0.9 in comparison to 0.5 – 0.8 depending on the environment), as well as visibly better training stability – for standard CRL, the correlation is visibly unstable through

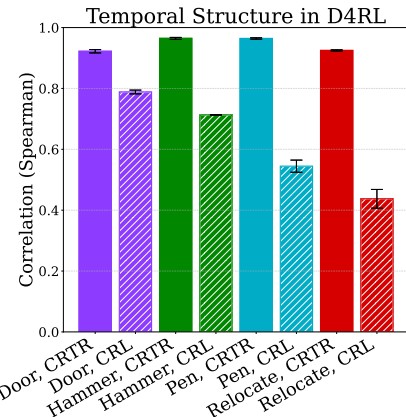

Figure 16: **CRTR improves temporal structure in robotics environments.** Comparison of Spearman's rank correlation metric for $CR^2$ (solid) and CRL (dashed) for D4RL offline datasets.

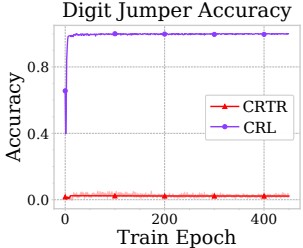 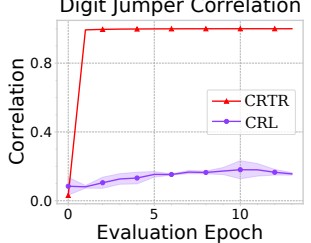 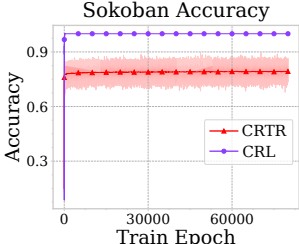

Figure 12: **In Digit Jumper, CRL quickly acquires near-perfect accuracy, however this is due to relying only on superficial features – the board layout.** Accuracy of classifying whether two states form a positive pair across the training, CRTR compared with CRL.

Figure 13: **In Digit Jumper, CRTR improves temporal structure in robotics environments.** Comparison of Spearman's rank correlation metric for $CR^2$ (solid) and CRL (dashed) for D4RL offline datasets.

Figure 14: **In Sokoban, CRL quickly acquires near-perfect accuracy, however this is due to relying only on superficial features, such as walls.** Accuracy of classifying whether two states form a positive pair across training: CRTR compared with CRL. The accuracy saturates at a value smaller than 1 for CRTR, as a result of containing in-trajectory negatives.

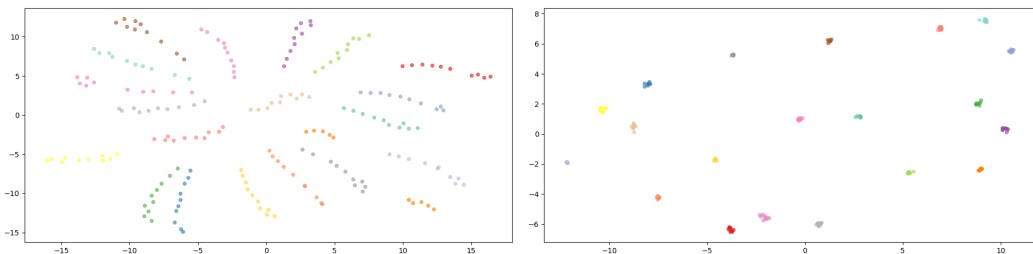

Figure 15: **CRTR makes representations reflect the structure of the combinatorial task.** t-SNE visualization of representations learned by CRTR (left) and CRL (right) for Digit Jumper. Colors correspond to trajectories. CRL representations (right) cluster within trajectories, making them useless for planning.

training and in some cases even becomes smaller as the training progresses. This result is a little surprising, and it is not fully clear why does the improvement happen. We hypothesize that this is because the initial position of the robot differs between trajectories and serves as a sort of slowly changing context, similarly to the Rubik's Cube case. We conclude that using CRTR results in a better temporal structure in the representation space for non-combinatorial problems.

## G   Correlation as a Measure of Representation Quality

To assess whether Spearman rank correlation is a reliable indicator of representation quality, we performed a grid of 96 short runs for each of three environments: Sokoban (12×12), Sokoban (16×16), and the Rubik's Cube. We varied four factors: network depth (8, 6, 4, 2), network width (1024, 16), representation dimension (64, 32, 16, 8), and the distance metric used in the contrastive loss (dot product, $\updownarrow_2$, $\updownarrow_2^2$).

Across all environments, the final Spearman correlation (computed with a budget of 1000 nodes) showed a strong relationship with the final success rate: 0.89 for 12×12 Sokoban, 0.80 for 16×16 Sokoban, and 0.90 for the Rubik's Cube. These results support the conclusion that Spearman rank correlation is a good measure of representation quality.

## H   Mutual Information Analysis

To estimate the conditional mutual information, we use NPEET package, which implements the method proposed in [13] that uses k-nearest neighbours for entropy estimation. We conduct the analysis using trajectories collected from the Sokoban or Digit Jumper environment, utilizing all

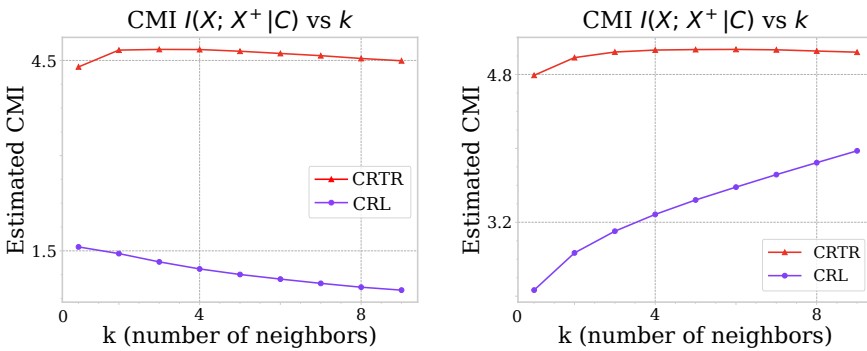

Figure 17: **CRTR optimizes the conditional mutual information while CRL does not, confirming out theoretical results 2.** Conditional mutual information estimated in Sokoban (left) and Digit Jumper (right) for representations learned by CRTR and CRL, for different values of nearest neighbors used for estimation.

---

**Algorithm 2** CRTR performs temporal contrastive learning, but samples negatives in a different way so that representations discard task-irrelevant context, boosting performance (See Fig. 2).

```
# dataset.shape == [num_traj, traj_len, obs_dim]
t0 = np.random.choice(dataset.shape[1], batch_size)
t1 = t0 + np.random.geometric(1 - discount, batch_size)
traj_id = np.random.choice(dataset.shape[0], batch_size)
# 1 new line of code for CRTR (our approach):
traj_id = np.repeat(traj_id[:batch_size // repetition_factor],
                    repetition_factor, axis=0)
batch = (dataset[traj_id, t0], dataset[traj_id, t1])
# further batch processing, the same for CRL and CRTR
```

---

transitions within these trajectories ($> 45k$ transitions for Sokoban and $> 20k$ for Digit Jumper). The variables used in the experiment are defined as follows:

- $X$: Current state embeddings, standardized using z-score normalization (mean 0, standard deviation 1) across the dataset. These embeddings are then projected onto a 3-dimensional subspace using Principal Component Analysis (PCA).
- $X^+$: Next state embeddings corresponding to transitions from $X$. The same standardization parameters and PCA transformation applied to $X$ are used for $X^+$ to ensure consistency.
- $C$: Trajectory identifiers (`traj_id`) encoded as 2-dimensional vectors sampled from a standard bivariate Gaussian distribution (i.e., $\mathcal{N}(0, I_2)$).

To mitigate the effects of the curse of dimensionality and ensure reliable performance of k-nearest neighbor (kNN)-based estimators, we reduce all high-dimensional representations to low-dimensional spaces (3D for state embeddings, 2D for trajectory identifiers). The conditional mutual information for CRTR and contrastive baseline is reported in Figure 17.

## I  Sampling Algorithm

Implementing our sampling algorithm requires changing just a few lines of code from prior temporal contrastive learning methods, as highlighted in Algorithm 2). The `repetition factor` governs the proportion of such negatives, thereby providing a controllable mechanism to interpolate between the standard and proposed objectives. Using data sampled in this way guarantees that some negative training pairs in each batch come from the same trajectory. We compare with potential alternative approaches in Appendix J.

## J  Ablations

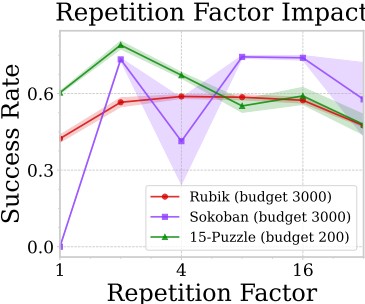

Figure 18: **A repetition factor of 2 consistently improves the performance.** Increasing the repetition factor for Sokoban, N-Puzzle, and Rubik's Cube, respectively.

**Repetition factor.** Our method introduces a single additional hyperparameter: the repetition factor $R$. This parameter controls the proportion of in-trajectory negatives and is critical for achieving strong performance. As shown in Figure 18, the impact of increasing $R$ varies by environment. For Sokoban, higher values of $R$ lead to only a slight decline in performance. In contrast, in many other environments, excessive repetition can significantly degrade results. While $R = 2$ is not always optimal, it consistently improves performance across all environments and serves as a strong default choice.

In Figure 19, we present detailed results showing how varying the repetition factor influences the success rate.

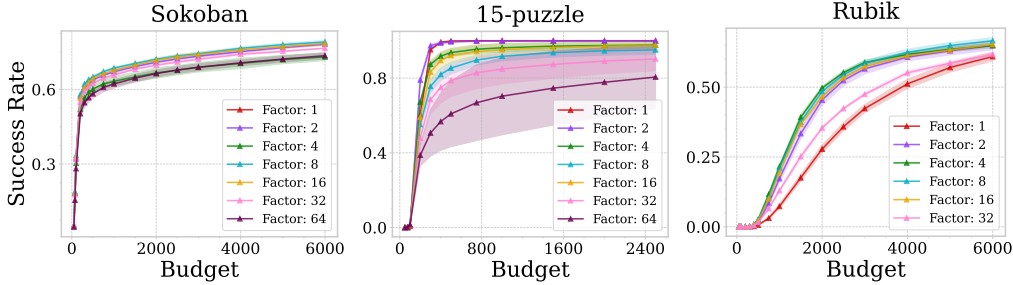

Figure 19: **Influence of the repetition factor depends on the environment type.** Increasing the repetition factor for Sokoban, N-Puzzle, and Rubik's Cube, respectively.

**Negatives.** We explored alternative methods for incorporating in-trajectory negatives into the contrastive loss. The first approach mimics the standard addition of hard negatives: given a batch $\mathcal{B} = (x_i, x_{i+})_{i \in \{1..B\}}$, we sample additional negatives $(x_{i-})_{i \in \{1..B\}}$, and compute the loss as

$$\mathcal{L} = \frac{1}{B} \sum_i \log \left( \frac{\exp\left(f(x_i, x_{i+})\right)}{\sum_{j \neq i} \exp(f(x_i, x_{j+})) + \exp(f(x_i, x_{i-}))} \right).$$

We considered three strategies for selecting in-trajectory negatives: sampling a state uniformly at random, choosing the first state, or choosing the last state of the trajectory. For Rubik's Cube, instead of choosing the last state—which is identical for all trajectories—we sample a random state farther from the solution to serve as a negative.

As shown in Figures 20 and 21, training with this approach did not yield strong performance. We hypothesized that the large prediction error introduced by the in-trajectory negatives $(x_{i-})$ caused

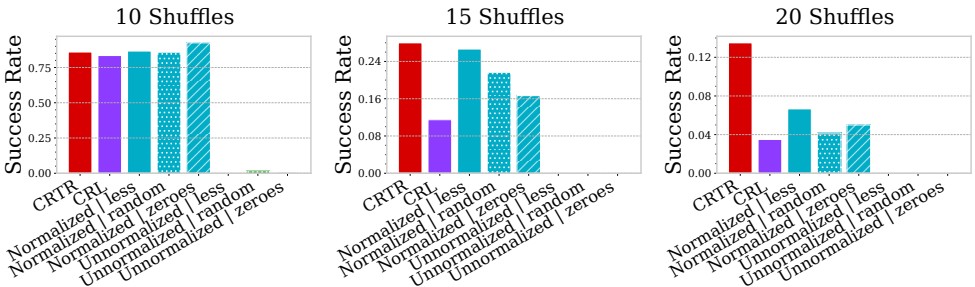

Figure 20: **In Rubik's Cube, CRTR outperforms all negative sampling strategies, when the number of scrambles increases.** Comparison of different methods for introducing in-trajectory negatives in the Rubik's Cube environment, with an increasing number of cube scrambles. While normalized negatives perform similarly to CRTR for a small number of scrambles, their performance deteriorates as the number of scrambles increases.

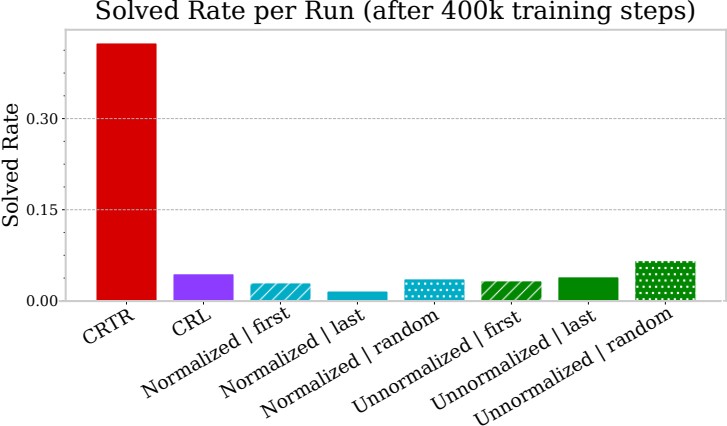

Figure 21: We compare various methods for introducing in-trajectory negatives in the Sokoban environment and find that only CRTR yields effective results.

excessively large gradients, destabilizing training. To mitigate this, we applied a normalization scheme: ensuring that the vector $[f(x_1, x_{1-}) \quad \cdots \quad f(x_B, x_{B-})]$ has the same Frobenius norm as the $B \times B$ matrix

$$\begin{bmatrix} f(x_1, x_{1+}) & f(x_1, x_{2+}) & \ldots & f(x_1, x_{B+}) \\ \vdots & \vdots & \ddots & \vdots \\ f(x_B, x_{1+}) & f(x_B, x_{2+}) & \ldots & f(x_B, x_{B+}) \end{bmatrix}.$$

This normalization enabled achieving comparable performance to CRTR on Rubik's Cube scrambled 10 times (Figure 20). However, CRTR still outperforms all negative sampling strategies on cubes scrambled 15 and 20 times.

For Sokoban, the only approach that consistently improved performance is CRTR, as demonstrated in Figure 21. We hypothesize that this is because removing contextual information is more challenging in Sokoban than in Rubik's Cube. In the latter, the context is more local and changes gradually over time, making it *softer*, while the context in Sokoban is constant throughout a trajectory. This is discussed in detail in Section 2.

While at first glance, repeating trajectories in a batch may seem equivalent to sampling in-trajectory hard negatives, the two approaches are different. In standard contrastive learning (as in CRL), an anchor $x$ pulls its positive $x_+$ closer and pushes negatives (e.g., $y_+$) away. However, negatives like $y_+$ are simultaneously pulled by their own anchors (e.g., $y$), which limits how far they are pushed by $x$. In contrast, when using in-trajectory negatives without anchoring them (e.g., $x$ pushes $x_-$ away, but $x_-$ has no anchor), these states can drift arbitrarily far in representation space. This is problematic, especially since in-trajectory negatives are harder (closer in structure), which results in stronger gradient updates. Our proposed method, CRTR, addresses this by anchoring all in-trajectory negatives. This keeps trajectories coherent and prevents such drift.

## K    Computational Resources

All training experiments were conducted using NVIDIA A100 GPUs and took between 5 and 48 hours each. The solving runs ranged from 10 minutes to 10 hours. In total, the project required approximately 30,000 GPU hours to complete.

## L    Things We Tried That Did Not Work

- Using separate encoders for future and present states did not improve performance.
- Adding extra layers to encode the action led to lower success rates.
- Using only in-trajectory negatives degraded performance.

- Modifying how current states are sampled in CRL (e.g., deviating from uniform sampling) did not yield improvements.

- Using A* solver with our representations could be greatly improved. Because distances in the latent space are only monotonically correlated—not linearly correlated—with actual distances, a modification to A* that would account for these discrepancies could bring huge gains.

- Distances between Rubik's Cube states, measured by the number of actions, almost always satisfy the triangle inequality with equality. Consequently, this metric cannot be faithfully embedded in Euclidean space, where equality in the triangle inequality occurs only for collinear points.

- Since Rubik's Cube actions are not commutative, a faithful Cayley graph structure could only emerge in a Euclidean space where vector addition is noncommutative—which would require a highly non-standard space.

