# OpenReview forum: "Contrastive Representations for Temporal Reasoning"
_NeurIPS.cc/2025/Workshop/UniReps — UniReps2025_

### Official Review · Reviewer_sCzp · 2025-09-10
**A good and interesting contribution**

**Confidence:** 2

**Review:**

Summary:

This paper identifies a critical failure mode in standard temporal contrastive learning, where models learn to rely on spurious, instance-specific "context" rather than the underlying temporal dynamics required for planning. To address this, the authors propose Contrastive Representations for Temporal Reasoning (CRTR), a novel method that uses a specific "in-trajectory" negative sampling scheme to force the model to learn context-invariant representations. The method is evaluated on several complex combinatorial reasoning tasks, where it demonstrates strong performance.

The central idea of "context" can be understood as the unique, static background or setup for a single instance of a puzzle. In the puzzle game Sokoban, for example, the context is the specific layout of walls and goal locations for a given level. This layout remains fixed for the entire duration of that puzzle, but is different for each new level the model encounters. The paper argues that standard learning methods use this static context as a "cheat sheet"; they learn to recognize the specific maze layout rather than learning the general, temporal skill of how to solve Sokoban puzzles. For a problem like the Rubik's Cube, which lacks a constant context, the initial scrambled state can serve a similar role as an "instance-specific" feature that simpler models might overfit to. CRTR is designed to ignore these instance-specific features and learn the general, transferable logic of the task itself.

A. Pros

A.1 Clear and Novel Concept:

The paper's core contribution is clear and well-motivated. The identification of the "context-dependency" problem in standard CRL is insightful, and the proposed solution of using in-trajectory negatives to make context an irrelevant feature is both elegant and effective. This provides a strong conceptual foundation and a novel direction for learning representations for temporal reasoning.

A.2 Benchmark Achievement on Rubik's Cube:

The empirical results are strong, with one finding being particularly noteworthy. The paper demonstrates that the learned representations can be used to solve 100% of arbitrary Rubik's Cube initial states using a simple greedy planning approach, without relying on an external search algorithm. This is a very impressive result that serves as a powerful demonstration of the quality of the learned representation space and stands as a significant benchmark for future work in this area. The emergence of sophisticated, human-like "block-building" strategies from this simple greedy method further underscores the depth of the learned representations.

B. Cons

While the conceptual idea and empirical results are strong, the paper's exposition of its theoretical framework and mathematical notation could be significantly improved to avoid potential confusion for the reader. The following areas would benefit from clarification:

B.1 Ambiguity in What is "Learned":

The notation max L(f) in the objective function is potentially confusing. It implies that the function f is being learned. However, if I understand the manuscript correctly, f is a composite function, consisting of a fixed, non-learned similarity metric (e.g., dot product) and a learned neural network encoder that produces the representations. I guess that the definition of f is as follows:

  f(s1, s2) = Similarity( Encoder(s1), Encoder(s2) ),

where the 1st Encoder and 2nd Encoder are the same function parameterized by a set of learnable parameters. Is my understanding correct? Regardless of whether it is correct or not, an explicit formulation that separates the learned encoder from the fixed metric would greatly improve clarity.

B.2 Relationship Between State and Embedding:

The paper introduces the raw state s_t (composed of context c and temporal part f_t) and the learned embedding X. However, the relationship X = Encoder(s_t) is not explicitly defined in the main text. A clearer definition would help readers understand that X is the output of a function applied to the raw state, and that the goal is to make X invariant to the context c.

B.3 Implicit Action Selection Mechanism:

The method for choosing actions is described as "greedily selecting the neighbor with minimum predicted distance". While effective, the paper would be stronger if it presented the explicit mathematical formula for this policy. If my understanding is correct, the policy might be defined as follows:

a* = argmax_{a ∈ A(s_t)} [ Similarity(Encoder(T(s_t, a)), Encoder(s_g)) ]

where T is a state transition function and s_g is the goal state. This would remove ambiguity and make the planning mechanism precise and self-contained.

C. Conclusion

Despite the need for clearer mathematical exposition, the paper's novel concept, strong empirical validation, and benchmark-setting results on the Rubik's Cube make it a valuable contribution to the field. The identified weaknesses can be addressed in a revision. Therefore, I recommend acceptance.

**Score:**

4

**Topic Fit:**

2

---

### Official Review · Reviewer_GGCm · 2025-09-10
**Good workshop paper**

**Confidence:** 3

**Review:**

**Summary:**

This paper aims to reduce reliance on search in combinatorial reasoning by learning better temporal representations. The authors argue that standard temporal contrastive learning tends to focus on static context shared by frames, failing to capture temporal dynamics. Their proposed method introduces an alternative contrastive sampling scheme designed to remove these spurious correlations and improve temporal reasoning. They evaluate across several domains (e.g., Sokoban, Lights Out, 15-puzzle, Rubik’s Cube) and report improved performance compared to both supervised baselines and BestFS, a state-of-the-art search algorithm.

**Pros:**

- The visualizations, particularly Figure 3, are very compelling. Figures overall are clear and informative.
- I appreciated the section on “things we tried that didn’t work”—this adds clarity and rigor.
- The analysis and motivation are well presented, and the problem addressed is an important one.
- Tackling temporal reasoning without explicit search is ambitious and relevant.

**Cons:**

-  If all elements in a batch come from the same trajectory, the method would fail (as shown in Han et al. 2025), but this limitation is not addressed.
- Temporal contrastive learning has a long history in self-supervised learning, yet very few related works are cited.
-  It is unclear whether CRL should be understood as temporal or spatial contrastive learning.
- The claim of “block-building emergence” for Rubik’s Cube seems overstated; it could simply be an artifact of training examples rather than a genuinely novel emergent strategy.

**Remarks**:

- The explanation of the greedy planning step (“picking the neighbor closest to the goal”) is vague and needs more detail.
- Figure 2: Why does the method perform worse in Sokoban compared to other domains?
- Reported success rate variance in Lights Out and 15-puzzle exceeds 1, which seems weird.
- There appears to be a missing logarithm in the lower bound on mutual information.

**Score:**

3

**Topic Fit:**

2

---

### Official Review · Reviewer_m31E · 2025-09-15
**Contrastive Representations for Temporal Reasoning**

**Confidence:** 4

**Review:**

Summary:

This paper proposes an algorithm to learn temporally structured representations in combinatorial reasoning tasks called Contrastive Representations for Temporal Reasoning (CRTR). They have tested this across five tasks: Sokoban, Rubik’s Cube, N-Puzzle, Lights Out, and Digit Jumper. The empirical evidence (t-SNE Visualization) shows that the standard temporal contrastive learning (CRL) is static in nature. In contrast, CRTR is temporal, and the representations capture the geometry of the environment (Sokoban environment) and don’t rely on contextual shortcuts. In the Rubik’s Cube experiment, the representations are learned like humans for solving, and it can solve problem instances of the cube in less than 6,000 moves (the average solve is 400 moves), without explicit planning. The proposed method solves the problem by understanding the task structure and balancing the global negatives with local negatives; thus, it enforces temporal consistency.

Pros:

++ The use of negative sampling to remove the irrelevant features is novel, and it has been shown to maintain the temporal consistency.

++ The steps and times required for the traditional search (Best-First Search) option to solve the cube moves are reduced by the learned representations, without any explicit reward.

++ The predictability of the learned representations in CRTR seems to hold for non-combinatorial problems as well.

Cons:

-- The human-like behaviour in learning the representation is not well-cited; a literature review of neuroscience papers that talk about representations would have added more context

-- The event boundary between one state to the next state in making the moves is not clear, and how it encodes those changes is not well-articulated.

**Score:**

4

**Topic Fit:**

3